# A Novel PDMS-Based Flexible Thermoelectric Generator Fabricated by Ag_2_Se and PEDOT:PSS/Multi-Walled Carbon Nanotubes with High Output Performance Optimized by Embedded Eutectic Gallium–Indium Electrodes

**DOI:** 10.3390/nano14060542

**Published:** 2024-03-20

**Authors:** Rui Guo, Weipeng Shi, Rui Guo, Chenyu Yang, Yi Chen, Yonghua Wang, Danfeng Cui, Dan Liu, Chenyang Xue

**Affiliations:** State Key Laboratory of Dynamic Measurement Technology, North University of China, Taiyuan 030051, China; nuc_guor0225@163.com (R.G.); swpjjt@163.com (W.S.); m18534715853@163.com (R.G.); yangwangcgsa@163.com (C.Y.); chenyi@nuc.edu.cn (Y.C.); wangyonghua@nuc.edu.cn (Y.W.); cuidanfeng@nuc.edu.cn (D.C.); xuechenyang@nuc.edu.cn (C.X.)

**Keywords:** FTEG, EGaIn, Ag_2_Se, PEDOT:PSS/MWCNTs

## Abstract

Flexible thermoelectric generators (FTEGs), which can overcome the energy supply limitations of wearable devices, have received considerable attention. However, the use of toxic Te-based materials and fracture-prone electrodes constrains the application of FTEGs. In this study, a novel Ag_2_Se and Poly (3,4-ethylene dioxythiophene): poly (styrene sulfonate) (PEDOT:PSS)/multi-walled carbon nanotube (MWCNT) FTEG with a high output performance and good flexibility is developed. The thermoelectric columns formulated in the work are environmentally friendly and reliable. The key enabler of this work is the use of embedded EGaIn electrodes, which increase the temperature difference collected by the thermoelectric column, thereby improving the FTEG output performance. Additionally, the embedded EGaIn electrodes could be directly printed on polydimethylsiloxane (PDMS) molds without wax paper, which simplifies the preparation process of FTEGs and enhances the fabrication efficiency. The FTEG with embedded electrodes exhibits the highest output power density of 25.83 μW/cm^2^ and the highest output power of 10.95 μW at Δ*T* = 15 K. The latter is 31.6% higher than that of silver-based FTEGs and 2.5% higher than that of covered EGaIn-based FTEGs. Moreover, the prepared FTEG has an excellent flexibility (>1500 bends) and output power stability (>30 days). At high humidity and high temperature, the prepared FTEG maintains good performance. These results demonstrate that the prepared FTEGs can be used as a stable and environmentally friendly energy supply for wearable devices.

## 1. Introduction

In recent years, the development of wearable devices has been constrained by the limited energy supply [1,2]. Piezoelectric nanogenerators [3], triboelectric nanogenerators [4], and solar generators [5] have been widely studied as power supply devices. However, they require light, continuous motion, and other demanding conditions for use. Thermoelectric generators (TEGs) can consistently and stably produce DC voltage from ambient temperature differences without strict environmental requirements [6,7], which is ideal for achieving a long battery life for wearable devices as well as self-powering capabilities. Therefore, developing and fabricating efficient and reliable TEGs is essential for advancing the technology of wearable devices.

Wearable flexible thermoelectric generators (FTEGs) have attracted increasing attention in recent years. The complete FTEG consists of n-type and p-type thermoelectric materials. However, most commercial FTEGs use toxic and low-content Te-based materials, which make the use of wearable devices environmentally unsustainable [8]. Therefore, Te-free materials such as carbon nanotubes, organic conductive polymers, and Te-free inorganic materials have become hot research topics in recent years. Ag_2_Se has gained significant attention as an n-type thermoelectric material with a high electrical conductivity and a low thermal conductivity near room temperature as well as for being environmentally friendly and for having abundant crustal reserves [9,10,11]. For p-type thermoelectric materials, the compounding of conductive polymers PEDOT:PSS and CNTs can improve the thermoelectric properties, which has also been studied [12]. SWCNTs hybridized with PEDOT:PSS greatly enhance the thermoelectric properties of composite films [13,14,15]. However, SWCNTs are costly. In contrast, MWCNTs are less costly and also improve the thermoelectric properties of composite films with PEDOT:PSS [16]. Thus, MWCNTs/PEDOT:PSS mixtures are ideal materials for wearable thermoelectric devices. However, FTEGs composed of Ag_2_Se and PEDOT:PSS/MWCNTs remains to be studied. On the device side, FTEGs can usually be classified into four types depending on their structure: π-type structure [17], Y-type structure [18], annular structure [19,20,21], and single-leg structure [22]. FTEGs with the Y-type structure exhibit uneven current and heat flow densities. The annular structure requires a specific shape for the heat source, and this structure is both difficult to fabricate in segments and challenging to manufacture. The single-leg structure leads to a high heat dissipation in the device, reducing the temperature difference and lowering the output power. The π-type structure can take full advantage of the properties of thermoelectric materials, that is, a high integration density and a simple preparation process. Thus, FTEGs with the π-type structure are promising candidates at present owing to their commercially viable structural design [23]. In addition, the electrode materials and manufacturing process are crucial factors for the stability and stretchability of flexible devices. Some FTEGs use solid metals to connect the thermoelectric columns in flexible substrates, such as silver [24,25] and copper [26,27]. Stress concentration during repeated bending can cause the rigid electrodes to fracture, making them unsuitable for wearable applications [28,29]. Liquid metals (LMs), such as eutectic indium gallium (EGaIn), which are in a liquid phase at room temperature, are infinitely deformable, exhibit good electrical and thermal conductivities, and have been studied for use as electrode materials [30,31]. For instance, Xu et al. [32] fabricated an FTEG using Ni–EGaIn-based covered electrodes, which produced 26.2 mV when worn and maintained an excellent performance under 30% tension deformation. Despite the high thermal conductivity of EGaIn, the two ends of the thermocouple cannot directly contact the hot and cold ends of the FTEG using covered electrodes, leading to a reduction in the obtained temperature difference [28]. In addition, most FTEGs adopt wax paper or other materials to assist the printing of EGaIn when fabricating electrodes, rendering the process very complex [31,33]. Therefore, developing a Te-free FTEG utilizing Ag_2_Se and organic–inorganic composites with an optimized electrode design is essential for energy harvesting and achieving a high-performance output for wearable thermoelectric devices.

In this work, a novel FTEG with a high output performance and good flexibility optimized by embedding EGaIn electrodes was developed. The thermoelectric columns were fabricated with Te-free Ag_2_Se and PEDOT:PSS/MWCNTs. The effects of different material weight ratios and fabrication pressures on the thermoelectric properties of the columns were investigated. In addition, the structure of the FTEG was simulated and designed. Furthermore, different FTEG prototypes with embedded EGaIn electrodes, Ag electrodes, and covered EGaIn electrodes were fabricated, enabling the comparison of their respective output performance. The embedded electrode design simplifies the fabrication process and endows the thermoelectric column with the ability to collect a larger temperature difference, thereby enhancing the FTEG output performance. The FTEG designed in this work is sustainable and environmentally friendly. Furthermore, it can provide a stable energy supply for extended periods as well as addressing the energy supply issue and lightweight requirement of wearable devices.

## 2. Materials and Methods

### 2.1. Materials

Ethanol, PEDOT:PSS was purchased from Shanghai Aladdin Industrial Co., Ltd. (Shanghai, China), polyvinylpyrrolidone (PVP, molecular weight: 40,000) was purchased from Wuxi Yatai United Chemical Co., Ltd. (Wuxi, China), Ag_2_Se was purchased from Hubei Xinhongli Chemical Co., Ltd. (Hubei, China), MWCNTs was purchased from Kona New Materials Co., PDMS was purchased from Suzhou Ruicai Semiconductor Co. (Suzhou, China), dimethyl sulfoxide (DMSO) was purchased from Shanghai Industrial Co., Ltd. (Shanghai, China), and EGaIn was purchased from Dongguan Metal Technology Co. (Dongguan, China). All reagents can be used directly without purification.

### 2.2. Fabrication and Characterization of the Thermoelectric Columns

Considering the wearable properties of FTEGs, n-type thermoelectric columns were realized using Ag_2_Se. PEDOT:PSS/MWCNT composites were selected for the p-type thermoelectric columns. Both materials are eco-friendly. The fabrication of the thermoelectric columns was performed using the cold pressing technique, as shown in Figure 1. First, using a pipette, put 1 g of PEDOT: PSS into a beaker and measure a volume ratio of 5% DMSO into the beaker. The mixed solution was mixed well with the ultrasonic shaker. A certain amount of MWCNT was added to the mixed solution and stirred evenly. Then, heating causes the mixed solution to become viscous. Finally, the viscous solution was poured into the mold for cold pressing to fabricate p-type thermoelectric columns. The thermoelectric column is cylindrical, with a diameter and height of 3 mm (Appendix A). According to the previous research of our group, the weight ratio of Ag_2_Se to PVP was determined to be 30:1 to fabricate n-type thermoelectric columns. The cold pressing process is the same as the p-type thermoelectric column. 

X-ray diffraction (XRD, DX-2700) measurements were conducted to determine the phase composition of the *n*/*p* thermoelectric columns. The surface morphology of the composite thermoelectric columns was observed via field-emission scanning electron microscopy (FESEM, SUPRA55 SAPPHIRE). Raman spectra were recorded using a Renishaw RM-1000 laser Raman microscope (Renishaw inVia Raman Microscope, New Mills, UK) at a laser wavelength of 532 nm. A test bench was designed to test the output performance of the thermoelectric columns. The temperatures of the hot and cold ends of the thermoelectric columns were set using a commercial Peltier module and a heating table. A commercial TCM-1030 temperature control module was used to control the temperature of the cold end. A digital multimeter (GDM-9061) was used to measure the output voltage of the thermoelectric columns. The variation in the output performance with temperature was obtained by setting the cold side temperature to 290 K and varying the hot side temperature from 295 to 305 K. The conductivity of the thermoelectric columns was measured using a four-probe semiconductor analyzer. 

### 2.3. Theoretical Simulations

Simulation can provide guidance for the preparation of FTEG. The commercial software COMSOL 5.6 is used to simulate FTEG. To study the influence of the leg length of the thermoelectric column on the output performance, heat transfer simulation and output performance simulation were carried out. In addition, the output performance of the embedded electrode and the covered electrode FTEG was simulated by controlling the length of the electrode. The effect of electrode length on the output performance of FTEG was studied by using embedded electrode FTEG.

### 2.4. Fabrication of the FTEGs

The fabrication process of the FTEGs is illustrated in Figure 2. (i) First, PDMS was fabricated by mixing the silicone base with a curing agent at a mass ratio of 10:1. The PDMS mixture was then poured into a mold created using computer numerical control (CNC) machining and cured under vacuum at 60 °C for 1 h. The PDMS molds were then filled with the thermoelectric columns and EGaIn electrodes (Appendix A). (ii) The fabricated n-type thermoelectric columns and p-type thermoelectric columns were alternately transferred to the PDMS molds through a vacuum pipette to form a complete circuit. (iii) The top side wall of the thermoelectric column was coated with silver paste. EGaIn was uniformly printed on the PDMS molds. It is crucial for the thermoelectric columns to be coated with metal since EGaIn with or without the oxide does not wet the columns, which would result in poor contact. (iv) Finally, a 5-mm-thick PDMS layer was encapsulated on both sides of the assembly. To facilitate the testing of the output performance of the FTEG, three FTEG prototypes were fabricated: one using Ag electrodes (referred to as t-FTEG), one using the covered EGaIn electrodes (referred to as l-FTEG), and one using the embedded EGaIn electrodes designed in this work (referred to as nl-FTEG). Each FTEG had three pairs of *n*/*p* thermoelectric columns. All fabricated prototypes had an area of 13 × 8 mm^2^ (Appendix A). 

### 2.5. Characterization of the FTEGs

The test system of FTEG is the same as that of the thermoelectric column performance test system (Appendix A). The voltage generated by the temperature difference (Δ*T*) was estimated as *V* = *nS_np_*Δ*T*, where *n* is the number of *n*/*p* thermocouples and *S_np_* is sum of the *S* of the *n*/*p* thermocouples. The output power (*P* = *U*^2^/*R*, where *U* is the output voltage and the *R* is the load resistance) is a key property in judging the performance of the device. Using a load resistor to match the internal resistance of the thermocouple array, the maximum output power was obtained when the load resistance was the same as the internal resistance.

## 3. Results

### 3.1. Material Properties

The performance characterization results of the thermoelectric columns are shown in Figure 3. Figure 3a shows the SEM image of the surface of the n-type thermoelectric column. The length of each Ag_2_Se nanorod (NR) is about 4.5 µm. Ag_2_Se nanorods overlap closely with each other, and the nanorods are bonded together by PVP. Figure 3b shows the SEM image of the p-type column. The MWCNTs are adhered through PEDOT:PSS and the material spacing was reduced by cold-pressing. Figure 3c shows the XRD pattern of the n-type column. The (112), (121), and (031) diffraction peaks of n-type are in good agreement with the PDF#24-1041 standard card of Ag_2_Se. The three XRD peaks are sharp, and their intensity is high. This indicates that Ag_2_Se is well crystallized. Furthermore, the fabricated n-type column has no obvious heterogeneous XRD peaks. The XRD patterns of the pure MWCNTs, PEDOT:PSS/DMSO, and p-type composite thermoelectric columns are shown in Figure 3d. The pure MWCNTs have a sharp (002) peak, indicating the crystalline and graphitic nature of the MWCNTs. The XRD pattern of PEDOT:PSS/DMSO does not contain any diffraction peaks, which indicates that it is in an amorphous state. The XRD pattern of the composite thermoelectric column reveals the presence of a clear characteristic peak at 25.9°, which corresponds to the (002) crystalline surface of the MWCNTs, without any impurity peaks. The XRD results show that the fabricated thermoelectric columns are free of significant impurities. Raman spectroscopy is often used to characterize the structural properties of polymers. Figure 3e shows the Raman spectra of the Ag_2_Se powder and the synthesized n-type thermoelectric column. The figure shows a peak at 132 cm^−1^, which is characteristic of Ag–Se bond formation [34]. The distinct Raman peak of the n-type thermoelectric column is consistent with the characteristic peak of the Ag_2_Se powder. The Raman spectra of the PEDOT:PSS/DMSO, MWCNTs, and PEDOT:PSS/MWCNTs are shown in Figure 3f. It can be seen that the characteristic peaks of the PEDOT:PSS/DMSO composite film are located at 1424.3 and 1568.8 cm^−1^. Compared with PEDOT:PSS (which has peaks at 1424.3 and 1572.6 cm^−1^ due to the symmetric stretching vibration of the benzoic acid structure in Cα = Cβ and the antisymmetric stretching of Cα = Cβ, respectively [35,36]), the peaks of the PEDOT:PSS/DMSO composite film are blue shifted, which is attributed to the conversion of the resonance structure of the PEDOT chain from the benzoic acid structure to the quinone structure after the DMSO treatment [37]. The MWCNTs have two distinct characteristic peaks at 1348 and 1582 cm^−1^, which correspond to the sp^3^ mode (D band) and sp^2^ mode (G band), respectively [13]. The characteristic peaks of the Raman spectra of the MWCNT/PEDOT:PSS composite column are located at 1348, 1424.3, 1568.8, and 1582 cm^−1^, which are consistent with the characteristic peaks of the MWCNTs and PEDOT:PSS/DMSO. Therefore, it can be concluded that the composite *n*/*p* thermoelectric columns are free of impurities.

The power factor (*PF*) of the thermoelectric columns is affected by their Seebeck coefficient (*S*) and conductivity (*σ*). The thermoelectric properties of *n*/*p* thermoelectric columns fabricated with different material mass ratios and fabrication pressures were invested (Appendix A). The power factor (*PF* = *S*^2^*σ*) of the thermoelectric column shows a nonlinear variation with the preparation pressure due to both *S* and *σ*. As shown in Figure 4a, n-type column reaches its maximum *PF* at a fabrication pressure of 10 MPa, namely 15.51 μW·m^−1^·K^−2^ at 305 K. Additionally, the *PF* of the p-type column reaches the maximum value of 10.44 μW·m^−1^·K^−2^ at 8 MPa at 305 K, as shown in Figure 4b. Considering the performance of the thermoelectric columns, the n-type columns prepared at 10 MPa and the p-type columns prepared at 8 MPa were selected for the FTEG fabrication.

### 3.2. Design and Optimization of FTEG

For the design of wearable FTEGs, two main aspects are considered in Figure 5. On the one hand, the equivalent circuit design is based on FTEGs in practical applications. The matching between the internal resistance and the external load resistance of wearable FTEGs is important for the practical application of FTEGs, and the output power (*P_L_*) of the FTEG will be maximum when the internal resistance of the FTEG is the same as the external resistance. On the other hand, the heat transfer between human skin and the FTEG and between the FTEG and air should be optimized. 

Figure 5a shows the equivalent circuit diagram of an FTEG working on human skin. The output power density of the device (*P_S_*, where *P_S_* = *P_L_*/*S*_0_, *S*_0_ = *L*·*D*, where *S*_0_ is the total area of the FTEG, *L* is the length of the FTEG, and *D* is the width of the FTEG) is determined by the number of thermoelectric pillars (*n*) and the fill factor (*f*, where *f* = *nS_g_*/*S*_0_ and *S_g_* is the area of a thermoelectric pillar). *n* and *f* are related to the thermoelectric column spacing (*l*). Each thermoelectric column in the FTEG generates a grain voltage (*V_g_*) and a grain resistance (*R_g_*). The goal of designing a mature and reliable FTEG is to maximize *P_s_* and reduce the material costs. The power per unit weight (*P_W_*) of wearable FTEGs is defined as the *P_L_* per gram of FTEG. In the dual-parameter model, the parameters *n* and *f* of the FTEG are optimized in order to optimize the power densities *P_S_* and *P_W_*. The load voltage of the device (*V_L_*) can be expressed as:(1)$$V_{L}=\frac{R_{L}\cdot V_{O}}{R_{L}+R_{teg}}=\frac{R_{L}\cdot V_{g}\cdot n}{R_{L}+R_{g}\cdot n}$$
(2)$$V_{0}=nS\Delta T$$
(3)$$n=\frac{L}{l_{ele}+d}\cdot \frac{D}{l_{ele}+d}=\frac{S_{0}}{{\left(l_{ele}+d\right)}^{2}}$$
where *V*_0_ is the open circuit voltage of the FTEG, *R_teg_* is its internal resistance, *R_L_* is the load resistance, *l_ele_* is the electrode length, and *d* is the diameter of the thermoelectric column. In addition, *P_S_* and *P_W_* are calculated as follows:(4)$$P_{s}=\frac{V_{L}^{2}}{R_{L}\cdot S_{\mathrm{teg}}}$$
(5)$$P_{W}=\frac{V_{L}^{2}}{R_{L}\cdot M_{0}}=\frac{V_{L}^{2}}{R_{L}\cdot n\cdot m_{0}}$$
where *M*_0_ denotes the weight of the FTEG, *S_teg_* is the total transverse area of whole thermoelectric columns, and *m*_0_ is the weight of a single thermoelectric column. Thus, the two-parameter model (*η*) can be formulated as: (6)$$\eta =P\frac{P_{S}}{P_{\mathrm{Smax}}}+\left(1-p\right)\frac{P_{W}}{P_{\mathrm{Wmax}}}$$
where *p* is the weighting factor, which ranges from 0 to 1 and *P_Smax_* and *P_Wmax_* are the maximum values of *P_S_* and *P_W_*, respectively. For simulations, the design should maximize *η*, i.e., maximize *P_S_* and minimize *P_W_*. Based on this model, the inter-column spacing, i.e., the filling factor, for optimizing the FTEG can be determined.

Figure 5b shows the results of the thermal model that was used to simulate the heat conduction between the body (*R_body_*), FTEG (*R_TEG_*), and air (*R_air_*). *T_body_* is the body core temperature, *T_air_* is the air temperature, *T_skin_* is the body skin temperature, and *T_cold_* is the temperature at the contact end between the FTEG and air. For the study of the thermal model, the resistances of the interfaces between the FTEG and the human body and air, as well as the resistance of thermally conductive components, such as heat sinks, are very small compared to the air resistance and the human body resistance. Thus, they can be safely ignored. The *V_L_* of the *R_L_* is associated with the thermal resistance according to:(7)$$V_{L}=\frac{R_{L}}{R_{L}+R_{teg}}nS\Delta T$$

The internal resistance of the device is related to the thermal resistance of the thermoelectric column. According to relevant reports, the maximum temperature difference of the thermoelectric column at maximum power depends on the thermoelectric materials and the leg length of the column (*l_teg_*) [38]. Therefore, *l_teg_* is a critical parameter in the design of FTEGs. It has been shown that the FTEG output power is maximum when *l_teg_* is optimized [39,40], which is expressed as:(8)$$l_{teg}=\kappa_{np}f(R_{body}+R_{air})$$
where *κ_np_* is the mean thermal conductivity of the two thermoelectric materials. The heat transfer by free convection in ambient air at 20 °C provides an air thermal resistance of about 0.1 m^2^·K·W^−1^. The body thermal resistance varies with the body location and was determined to be about 0.02 m^2^·K·W^−1^ on the wrist [41]. Based on this model, it is possible to optimize the leg lengths of the thermoelectric columns. 

Both models illustrate that the leg length and inter-column spacing of the thermoelectric columns significantly affect the output performance of the FTEGs. The corresponding simulation was carried out using the COMSOL software 5.6. Figure 5c shows the simulation results for a single thermoelectric column. Figure 5d shows the simulation results regarding the effect of different electrode lengths on the FTEG output. Figure 5e shows that the output voltage increases as *l_teg_* increases. However, the increase in leg length leads to an increase in the internal resistance of the thermoelectric column. In addition, to increase comfort during actual use, 3-mm-long thermoelectric columns were selected. Figure 5f shows that the FTEG generated the highest *P_s_* and *P_W_* when the electrode length was 4.5 mm. This means that the inter-column spacing of the thermoelectric columns was 1.5 mm. The covered EGaIn-based electrode and embedded electrode were also simulated using the COMSOL software 5.6. Figure 5g shows the simulated temperature difference of the FTEG thermoelectric columns. The Δ*T* obtained for the nl-FTEG is 14.93 K, while that obtained for the l-FTEG is 14.45 K, which is 2.5% lower than the former. Figure 5h shows the simulated output voltage of the FTEG. The output voltage of the nl-FTEG is 0.42 mV, while that of the l-FTEG is 0.40 mV. The output voltage is linearly related to the temperature difference. The simulation results in this study demonstrate that the optimized embedded electrodes enhance the capability of the thermoelectric columns to collect the temperature difference. The improved electrodes enable close contact between the thermoelectric column and the hot and cold sides of the system. This leads to a higher temperature gradient across the thermoelectric columns, which improves the device performance and increases power generation. The optimized electrodes are crucial in maximizing the ability of the thermoelectric columns to collect the temperature difference and enhancing the overall efficiency of the thermoelectric system.

### 3.3. FTEG Performance Characterization

The test results of the thermoelectric performance of the FTEG are shown in Figure 6. Figure 6a displays the equivalent circuit of the test system. Figure 6b shows the relationship between the output voltage and the temperature when the FTEG is not encapsulated. The output voltage of the FTEG has an excellent linear relationship with temperature. The *S_np_* of each pair of *n*/*p* pillars was calculated to be 65.53 μV/K, which matches the test results for a single pillar. Figure 6c shows the output power test results of the t-FTEG. The output power of the t-FTEG is maximum at 305 K. After further testing, it was found that the t-FTEG has the maximum output power (8.31 μW) at a load resistance of 17 Ω. The internal resistance is inconsistent with the expected data. This discrepancy may be caused by (i) the fact that the silver–semiconductor contact resistance was not optimized, (ii) the mechanical fracture of the solid silver electrode, and (iii) the unavoidable surface oxidation of the semiconductor pillar prior to the deposition of the silver electrode. A high resistance leads to a low output power. Figure 6d,e show the output power test results for the l-FTEG and nl-FTEG. At 305 K, the maximum output power of the l-FTEG is 10.68 μW, which is 29.1% higher than that of the t-FTEG. The output power of the nl-FTEG is 10.95 μW, which is 2.5% higher than that of the l-FTEG and 31.6% higher than that of the t-FTEG. The experimental results deviate slightly from the simulation results. A possible reason for such a deviation may lie in external factors or variables in the experiment that were not accounted for in the simulation. These factors could include variations in temperature, humidity, or other environmental conditions that can affect the device performance and increase its internal resistance. Additionally, manufacturing tolerances or inconsistencies in the actual device could contribute to the observed deviation from the simulation result. The maximum output power density of the nl-FTEG is 25.83 μW/cm^2^ at Δ*T* = 15 K. Table 1 shows that the nl-FTEG prepared in this work exhibits a superior output performance. Compared with previously studied TEGs, the nl-FTEG uses less material and has a higher output power density. The FTEG was subjected to cyclic bending strain to determine whether it can operate effectively under repeated use as an energy harvester without electrical or mechanical degradation (Appendix A). After 1500 bending cycles, the t-TEG showed a significant decrease in conductivity, namely 80% of the original conductivity, as shown in Figure 6f. The fracture of the solid Ag electrode in the t-FTEG leads to an increase in the internal resistance of the device. The nl-FTEG, on the other hand, had only a 3.3% reduction in conductivity due to the excellent performance of the embedded EGaIn electrode.

Considering the practical use of the device, an output stability test was carried out, and the corresponding results are shown in Figure 7a. The output power of the nl-FTEG was tested every 2 days. It was found that the nl-FTEG can maintain good stability for a long time. After 30 days, the output power of the nl-FTEG decreased. The most probable reason for this is that a small amount of air entered the device during the PDMS encapsulation process, causing oxidation of the thermoelectric columns and increasing the internal resistance of the device. The device maintained excellent stability, with the output power of the nl-FTEG being 10.11 μW after 30 days, which represents a reduction of only 8.31%. The PDMS packaging process could be improved in subsequent experiments, and it is envisaged that it will be possible for the nl-FTEG to become more stable. The resistance change of nl-FTEG was tested at different temperatures. In Figure 7b, as the temperature increases, the resistivity of Ag_2_Se and PEDOT: PSS/MWCNTs decreases with increasing temperature, resulting in a decrease in the resistance of nl-FTEG. In the high humidity environment, the resistance of nl-FTEG did not change significantly. As shown in Figure 7c, the output voltage of nl-FTEG did not change significantly with the increase of humidity, which will not affect the thermoelectric conversion efficiency. As shown in Figure 7d, it is confirmed that the PDMS package is very good, and the increase in humidity will not cause the water molecules inside the nl-FTEG to adsorb on the surface of the material or penetrate into the internal structure, thus having little effect on the performance of the nl-FTEG. However, in a humid environment, the electrodes of nl-FTEG may be oxidized or corroded, resulting in a decrease in battery performance. To conclude, the prepared nl-FTEG exhibits excellent strength and long-term serviceability, reliable power supply, and excellent output performance, making it an ideal energy supply for wearable devices.

## 4. Conclusions

In summary, a novel Ag_2_Se and PEDOT:PSS/MWCNT FTEG with embedded EGaIn electrodes with a high output performance was fabricated. The embedded electrodes permit the direct printing of EGaIn onto the PDMS mold, thereby simplifying the fabrication process of FTEGs and enabling the collection of larger temperature differences by the thermoelectric column. Three different FTEG prototypes were prepared and characterized. The nl-FTEG had a maximum output power of 10.95 μW measured at Δ*T* = 15 K and an output power density of 25.83 μW/cm^2^. The output power of the nl-FTEG was 31.6% higher than that of the t-FTEG and 2.5% higher than that of the l-FTEG. Even after 1500 bends, the nl-FTEG showed only a 3.3% reduction in conductivity, which highlights its excellent flexibility due to the use of the flexible EGaIn electrodes. The FTEG designed in this work offers a long-term stable energy supply and represents an ideal and environmentally friendly solution for powering wearable devices.

## Figures and Tables

**Figure 1 nanomaterials-14-00542-f001:**
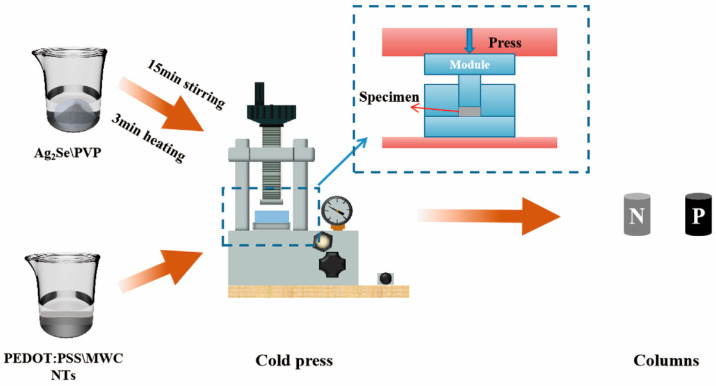
Schematic diagram of cold-pressed fabrication of thermoelectric columns.

**Figure 2 nanomaterials-14-00542-f002:**
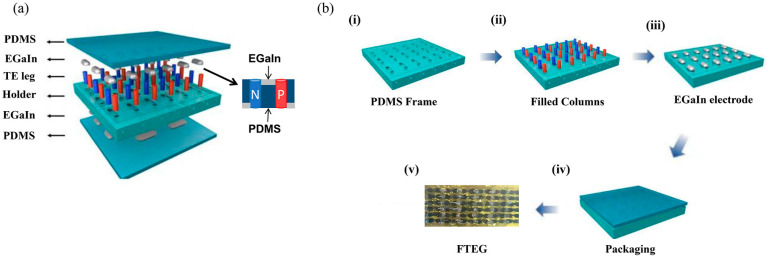
(**a**) The overall schematic of FTEG, (**b**) the fabrication process for FTEG.

**Figure 3 nanomaterials-14-00542-f003:**
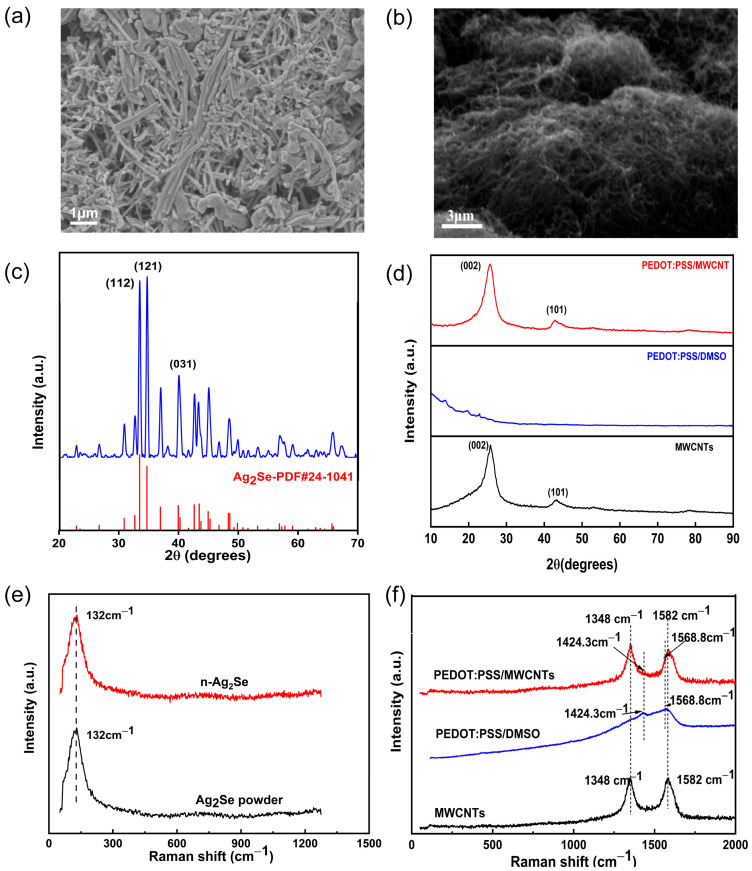
The SEM images of (**a**) n-type and (**b**) p-type, the XRD patterns of (**c**) n-type and (**d**) p-type, (**e**) Raman spectrum of Ag_2_Se powder and n-type thermoelectric columns, (**f**) Raman spectrum of MWCNTs, PEDOT:PSS/DMSO, and PEDOT:PSS/MWCNTs thermoelectric columns.

**Figure 4 nanomaterials-14-00542-f004:**
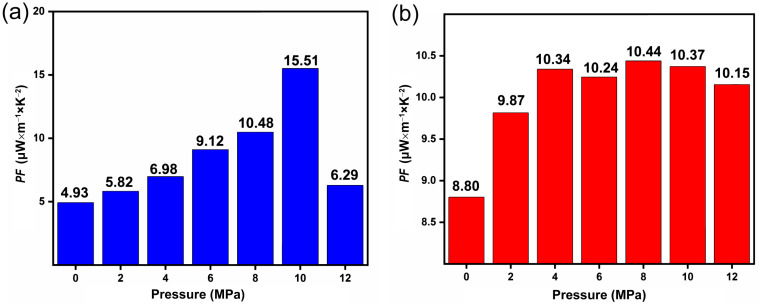
(**a**) The *PF* results of n-type column with different fabrication pressures, and (**b**) *PF* results of p-type column with different fabrication pressures.

**Figure 5 nanomaterials-14-00542-f005:**
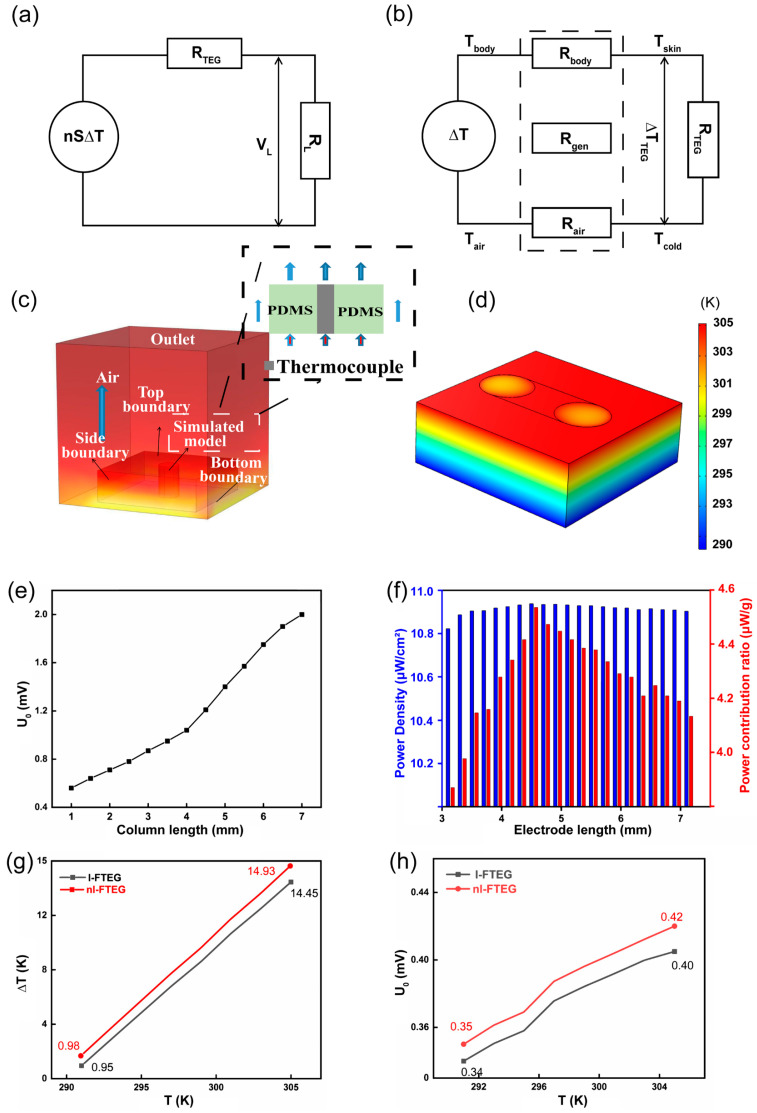
(**a**) Electric model of TEG in contact with human skin, (**b**) thermal model, (**c**) Simulation of heat transfer for leg length, (**d**) simulation of heat transfer for electrode length, (**e**) simulation results of the effect of different leg lengths on output voltage, (**f**) effect of electrode length on output density and power per unit weight, (**g**) simulation results of temperature difference actually obtained for different electrode, (**h**) simulation results of output voltage for different electrode.

**Figure 6 nanomaterials-14-00542-f006:**
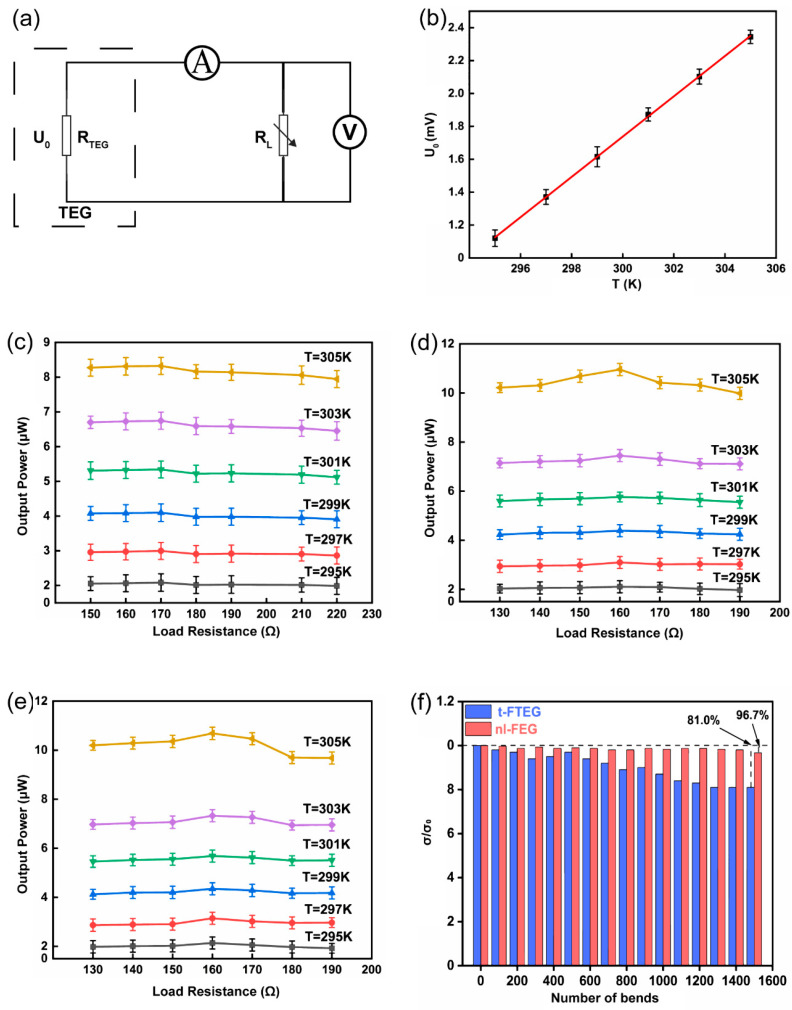
(**a**) Test schematic, (**b**) FTEG output voltage versus temperature, (**c**) output power of t-FTEG versus load resistance at different temperatures, (**d**) output power of l-FTEG versus load resistance at different temperatures, (**e**) output power of nl-FTEG versus load resistance at different temperatures, (**f**) bending test of t-FTEG and nl-FTEG.

**Figure 7 nanomaterials-14-00542-f007:**
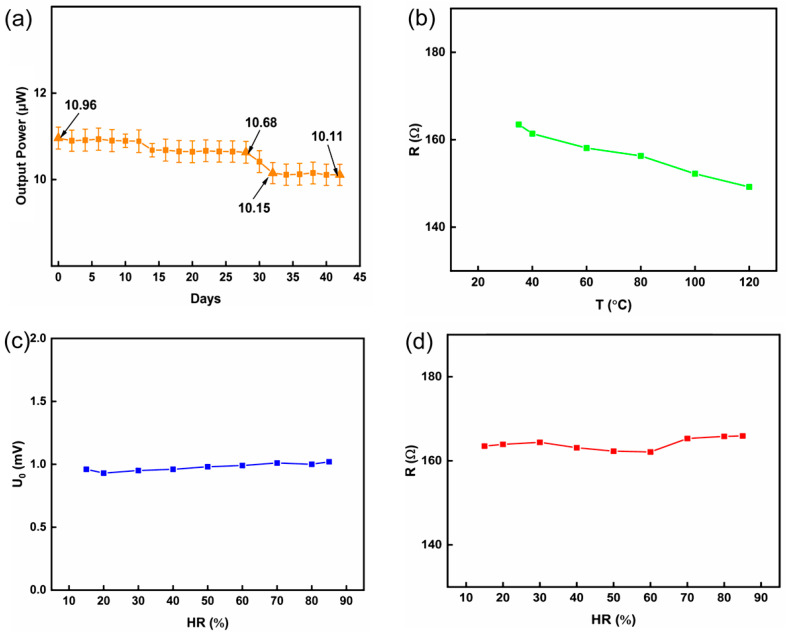
(**a**) stability test results of nl-FTEG, (**b**) the resistance change of nl-FTEG at different temperatures, (**c**) the change of output voltage of nl-FTEG under different humidity, (**d**) the change of resistance of nl-FTEG under different humidity.

**Table 1 nanomaterials-14-00542-t001:** Comparison of output power density with other FTEG.

Materials	Process	Electrode	Number	Power-Density (μW/cm^2^)	Ref.
Bi-Te compounds	bulk	EGaIn	30	8.32	[42]
polyurethane (PU), graphene-composites	bulk	Semi-LM	100	7.3	[30]
Bi_0.5_Sb_1.5_Te_3_, Bi_2_Se_0.3_Te_2.7_	bulk	EGaIn	64	5.6	[33]
Ag_2_Se	film	Ag	6	8.0	[9]
Ag_2_Se	film	cooper	20	1.43	[43]
Ag_2_Te, PEDOT:PSS	nylon	Ag	4	0.6	[44]
Ag_2_Se, PEDOT:PSS/MWCNTs	bulk	EGaIn	6	25.83	This work

## Data Availability

Data are contained within the article and Appendix A.

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
