# Peer review of "A Novel PDMS-Based Flexible Thermoelectric Generator Fabricated by Ag2Se and PEDOT:PSS/Multi-Walled Carbon Nanotubes with High Output Performance Optimized by Embedded Eutectic Gallium–Indium Electrodes"

_nanomaterials, 2024, doi:10.3390/nano14060542_

Round 1
Reviewer 1 Report
Comments and Suggestions for Authors
The manuscript is well written, the language is understandable, and the reader has no problems with receiving the presented information. In the case of Figure 3, it should be divided into several separate ones regarding the discussed methods and placed in the text so that they are closer to the discussed issue.
Author Response
Dear Editor and Reviewers,
Thanks for the reviewers' comments and suggestions regarding our manuscript entitled "A Novel PDMS-based Flexible Thermoelectric Generator Fabricated by Ag2Se and PEDOT:PSS/multi-walled carbon nanotubes with High Output Performance Optimized by Embedded Eutectic Gallium–Indium Electrodes" (ID: nanomaterials-2899203). Those comments and suggestions are valuable and very helpful for revising and improving our paper and the important guiding significance to our research. All comments and suggestions have been replied to point-by-point below, and revised portions are marked in yellow in the revised manuscript. We resubmit the revised manuscript and thank you for your kind consideration.

Reviewer 2 Report
Comments and Suggestions for Authors
In this paper, the authors prepared a π-type module with silicone rubber using PEDOT:PSS and MWCNT composite as p-type semiconductors and Ag2Se as n-type semiconductor. However, their performance is low and lacks practicality. In addition, there is no connection to the theme of "Nanomaterials" in this journal. Therefore, the reviewers reject this paper for publication in this journal.
If the publisher decides to publish this paper in this journal, the authors should at least revise the following points;
(1) This paper uses MWCNTs as the CNTs used to form a composite with PEDOT:PSS. However, in general, there are many reports showing that SWCNTs have better thermoelectric properties. (Org. Electron., 51, 304 (2017); ACS Appl. Mater. Interfaces, 13, 12131 (2021); Compos. Commun., 27, 100897 (2021))
The authors should explain why they chose MWCNTs instead of SWCNTs, citing the above references.
(2) In this paper, the composite ratio of PEDOT:PSS and MWCNTs was only investigated in the range of 0-20 wt% MWCNTs; when SWCNTs were used, the highest thermoelectric conversion properties were found at 75-80 wt% SWCNTs. The authors should perform measurements in the range of 0-100 wt% composite ratio. Also, what is the rationale for the maximum MWCNT composite ratio of 10 wt%?
The composites are also fabricated under different pressure conditions. Clearly state the cause of the change in thermoelectric properties at different pressure and explain why the maximum value is obtained at 10 MPa.
(3) The resolution of the figure is too poor and difficult to read. Especially for Figs. 1, 2, S1, S7, and S8, data with higher resolution should be pasted. Among others, the resolution of the photos is too poor and it is not clear what they show.
Author Response

(The authors gave the same response as above.)

Reviewer 3 Report
Comments and Suggestions for Authors
The paper "A Novel PDMS-based Flexible Thermoelectric Generator Fabricated by Ag2Se and PEDOT:PSS/multi-walled carbon nanotubes with High Output Performance Optimized by Embedded Eutectic Gallium–Indium Electrodes" proposes a novel flexible thermoelectric generator (FTEG) designed for wearable technology applications. The study addresses an interesting topic, and presents a detailed and systematic approach to the fabrication and characterization of the device. In my opinion it can be accepted with some minor revions.
My main concerns regards the choice of the material that needs to be justified in detail in the introduction.
I am not qualified to comment on the quality of english language
Author Response

(The authors gave the same response as above.)

Reviewer 4 Report
Comments and Suggestions for Authors
Please see the attached file

Author Response

(The authors gave the same response as above.)

Round 2
Reviewer 2 Report
Comments and Suggestions for Authors
The authors explained to the reviewer that they used MWCNTs because they are less expensive than SWCNTs, but it is not clear where this point is indicated in the revised manuscript, so readers cannot understand why they used MWCNTs with poor performance.
The authors should clearly explain the significance of this paper by stating the price advantage of MWCNTs after explaining with references suggested by the reviewer that the composite of SWCNTs and PEDOT:PSS is superior to the composite of MWCNTs and PEDOT:PSS in terms of thermoelectric performance.
(Suggested references)
1) Org. Electron., 51, 304 (2017).
2) ACS Appl. Mater. Interfaces, 13, 12131 (2021).
3) Compos. Commun., 27, 100897 (2021).
The authors explain that as the percentage of MWCNTs in the composite increases, the Seebeck coefficient of the composite film decreases due to the higher carrier density of the MWCNTs. But if this is correct, why does the Seebeck coefficient not decrease monotonically as the percentage of MWCNTs increases?
If the carrier density is really related, then the carrier density should be quantitatively estimated in some way and the correlation between the carrier density and Seebeck coefficient should be investigated. If the authors do not have any technique to quantify the carrier density, they should ask researchers who analyze the correlation between carrier density and Seebeck coefficient of organic thermoelectric conversion materials to measure it. Otherwise, the reviewer concludes that this paper should not be published because of the inadequate discussion of this part of the paper.
Author Response
Thanks for the reviewers' comments and suggestions regarding our manuscript entitled "A Novel PDMS-based Flexible Thermoelectric Generator Fabricated by Ag2Se and PEDOT:PSS/multi-walled carbon nanotubes with High Output Performance Optimized by Embedded Eutectic Gallium–Indium Electrodes" (ID: nanomaterials-2899203). Those comments and suggestions are valuable and very helpful for revising and improving our paper and the important guiding significance to our research. All comments and suggestions have been replied to point-by-point below, and revised portions are marked in yellow in the revised manuscript. We resubmit the revised manuscript and thank you for your kind consideration.

Reviewer 4 Report
Comments and Suggestions for Authors
The authors made improvements on the manuscript according to reviewer's suggestions. There is a single minor thing, that authors should address, before the manuscript becomes suitable for publication
1. There is still problem with font sizes into some Figures. In particular, in Figure 2, Figure 3c, 3d, 3e and 3f, as well as Fig 5c, font size into Figures must be enlarged, even more. They cannot be read when printed.
Author Response

(The authors gave the same response as above.)

Round 3
Reviewer 2 Report
Comments and Suggestions for Authors
The authors well-responded to the reviewer's comments, so the reviewer concluded that this paper will be acceptable for the publication in the journal, Nanomaterials.